


# Evaluation of the resilience of fishery ports to typhoons: a case study on Dongsha fishery port

Yachao Zhang[1], Xiaojie Zhang[1], Jufei Qiu[1], Aifeng Tao[2], Yanfen Deng[1], Jianli Zhao[1], Jianfeng Wang[1], Wentao Huang[3]

[1]East Sea Marine Environmental Investigation & Surveying Center, SOA China, Shanghai 200137, China

[2]Key Laboratory of Ministry of Education for Coastal Disaster and Protection, Hohai University, Nanjing 210098, China

[3]Shanghai East Sea Marine Engineering Survey & Design Institute, Shanghai 200137, China

*Correspondence to:* Jufei Qui (qjf@ecs.mnr.gov.cn)

**Abstract.** After standard seawalls have been built successfully, fishery ports become the structures most easily damaged during a typhoon. Assessments of the resilience of fishery ports to typhoon damage would be useful for identifying weaknesses and implementing corrective measures to protect fishing boats from a typhoon. This study describes a versatile methodology for conducting this type of quantitative assessment at fishery ports. The Dongsha fishery port in Zhejiang Province was selected as a case study to test the results derived from a high-precision Hydrodynamic Flexible Mesh model coupled with the Spectral Wave model. First, typhoon characteristics were assessed based on historical typhoons in the study area, and then, the wind, tide, storm surge, and waves were modeled and tide-surge interactions were investigated. Through comparisons of the destructive parameters from the typhoon assessment with the design and structural parameters of the fishery port, the resistance level of the Dongsha fishery port against typhoons was determined to be 12, and the main weaknesses of the port's defenses were found to be located near feature points T2, T3, T8, and T15. The results obtained demonstrate that the proposed methodology can be used to acquire valuable information on the resilience of fishery ports to typhoons.

## 1 Introduction

As one of the countries with the largest fishery resources around the world, China ranks first in the world in terms of the output of aquatic products, number of fishing boats, and number of fisheries employees (NDRC and MARA, 2018). However, as China lies on the west coast of the Pacific Ocean, its coastal areas are susceptible to various marine disasters, especially typhoons and storm surges (Ministry of Natural Resources of the People's Republic of China, 2019). Within China, Zhejiang Province near the East China Sea is well-known for fishing. The total marine fishery production output here was ranked first nationally at 3,200,000 t. Coastal areas within Zhejiang, especially the city of Wenzhou, are vulnerable to typhoon-related damage (Du et al., 2020; Shi et al., 2020b). Almost every year, more than one typhoon strikes the coast of Zhejiang Province, and these typhoons frequently cause damage to the breakwater structures, wharfs, and fishing boats. According to the Zhejiang Marine Disaster Bulletin (2019), a total of 2064 fishing boats were damaged by typhoons, and the direct economic losses due to typhoons amounted to 87.25 hundred million yuan (Department of Natural Resources of Zhejiang Province, 2019). Since record keeping began, the largest storm surge event near the Dongsha fishery port occurred in 1997 (210 cm at the Kanmen tide gauge station). Significant fluctuations in the sea level are caused by the strong winds in the low-pressure storm systems that cross over the Dongsha fishery port. As storms pass over the sea, the conditions create storm surges. Low atmospheric pressure and winds cause an increase in water levels at nearby coastal areas, which often leads to flooding (Wang et al., 2017). After standard seawalls are successfully built, disaster prevention and mitigation efforts at fishery ports become particularly important. Knowledge of the degree of resilience of fishery ports to typhoons would be of great benefit to disaster prevention plans and coordination of mitigation activities within a region.

Most research on fishery ports has focused on the biology and ecology of a port and its geomorphic stability. However, the ability of fishery ports to resist the damage caused by typhoons has not received much research attention yet. Notably,


Premwadee et al. (2006) studied the trends in marine fish catches at the Pattani fishery port, and Kawaguchi et al. (1995) presented construction recommendations for an offshore fishery port to prevent coastal erosion following hydraulic model tests and numerical simulations of wave induced currents near the port. Additionally, there have been numerous studies about the risks of hurricanes or typhoons at home and abroad. In America, the National Weather Service storm surge model, named SLOSH (Sea, Lake, and Overland Surge from Hurricanes), has been used to delineate coastal areas susceptible to hurricane storm surge flooding (Glahn et al., 2009). A computer simulation of super typhoon Haiyanin with the resulting wave heights and storm surge levels was made using the MIKE21 model in Tacloban city (Prelligera et al., 2014). Li et al. (2020) examined the dependence of typhoon-induced storm surge and wave setup effects on the typhoon intensity and size. MIKE21 was also used to evaluate the overtopping risk of seawalls and levees from the combined effects of the storm tide, sea level rise, and land subsidence in Shanghai (Wang et al., 2011). A methodology for storm surge risk assessments in coastal counties was established following research in Jinshan District, Shanghai city (Shi et al., 2020a).

Estimating the resilience of fishery ports to typhoons is a difficult task. In particular, because each fishery port is different in terms of its geographical location, topography, anchoring water, and shape, we cannot carry out one assessment under the same typhoon conditions or even for completely different typhoons. Additionally, the storm surge can be influenced significantly by the landfall location of a typhoon with the same pressure (Sun et al., 2015). Abeshima et al. (2017) clarified the mechanism of port disturbance generation at the Kumaishi fishery port and concluded that the quantitative indicator $H_{1/3}$ (over 2.0 m) can be introduced as a decision indicator for evacuations by observational statistics. Some exploratory work has been conducted in China on fishery ports' resistance to damage caused by typhoons. Notably, one study used an analytical hierarchy process for indexes of wind and wave features, the number of sheltering boats, anchoring methods, emergency measures, and the local management system to assess the relative preparedness of fishery ports in Xiamen against typhoons (Dongshui and Qionglin, 2019). Based on the nested model of Delft3D, the fishery ports were first evaluated in terms of the following three aspects: the level of shoreline facilities, anchorage areas, and breakwaters to lessen typhoon impacts. However, the maximum observed frequency of the wind direction was roughly regarded as the typhoon pathway, which was the key factor in that study. Importantly, the interaction of the tide and surge was not taken into account.

This study describes a systematic and quantitative method for assessing the resilience of fishery ports to typhoons. The method can be used to conduct comparisons among different fishery ports, and the proposed method also relies on basic data for the three aspects described above. Additionally, the typhoon resistance capability of a fishery port is indicated by the sustainable maximum wind scale of the port. Meanwhile, typhoon pathways and tide-surge interactions, the key factors of the assessment, are studied in detail. After deriving a quantitative value for the resistance level of a fishery port against typhoons, effective countermeasures for typhoons can be proposed, and such data should also be useful for making judgments as to the need for evacuations by administrators.

## 2 Materials and methods

### 2.1 Study area

The Dongsha fishery port is located on the east side of Dongtou Island (121°10′–121°11′E and 27°50′–27°51′N) in the city of Wenzhou, China (Fig. 1). It is C-shaped and surrounded on three sides by mountains, which makes it a natural sheltered harbor for fishing boats. Presently, it is the best sheltered harbor in Wenzhou. The length of the fishery port coast is 5.17 km, and there is a 0.35 km long breakwater, which was built at the entrance of the fishery port. The water area of the port is approximately 750,000 m² with a depth of 3–9 m.

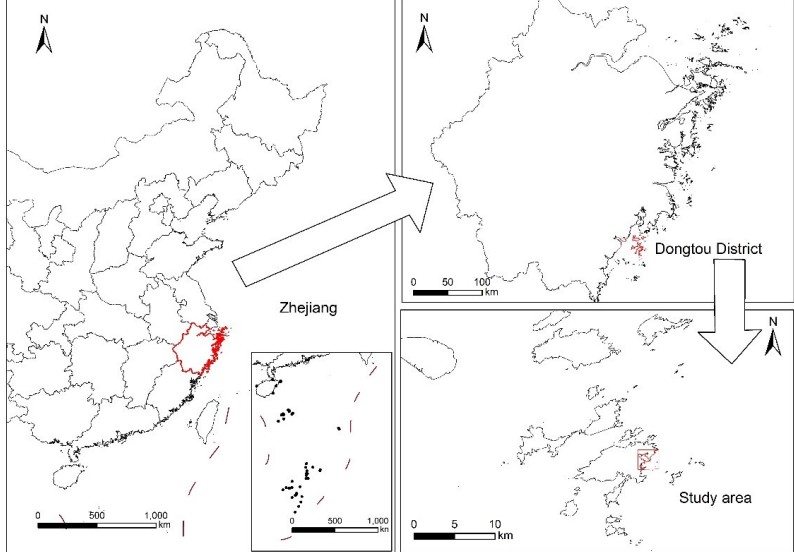

**Figure 1.** Study area in Dongsha, Wenzhou, China.
**2.2 Data**
To model a specific area of a vulnerable fishery port, accurate topographic, meteorological, and other types of basic data are
required. Here, multisource data were classified into four types (Table 1) and were used to run and validate the Hydrodynamic
Flexible Mesh (HD FM) model coupled with the Spectral Wave (SW) model for the Dongsha fishery port. The topographic
data, which were at the same datum plane, were collected to construct the numerical model, and the meteorological data and
hydrologic data were used as dynamic data and validation data for the numerical model. The design and structural parameters
of the Dongsha fishery port were compared with the numerical simulation results, and then, these results were used to judge
the resistance of the port.
**Table 1.** Multisource data used to perform and validate the model.

| Data type | Element | Time series | Description | Source |
|---|---|---|---|---|
| Meteorological data | Wind | 1961–2015 | Wind velocity and direction | Wenzhou Marine Environmental Monitoring Center |
| | Historical typhoon records | 1949–2018 | Time, location, and intensity of each typhoon track point | China Meteorological Administration |
| | Tide | 2014.10 | Hourly tidal level | Wenzhou Marine Environmental Monitoring Center |
| Hydrological data | Storm surge | 1997–2015 | \ | Wenzhou Marine Environmental Monitoring Center |
| | Current | 2014.10 | Hourly flow velocity and direction | Actual measurement |
| | Wave | 1997–2015 | Significant wave height | Wenzhou Marine Environmental Monitoring Center |
| Topographical data | Topography | 2016.1 | Depth of fishery port and chart | Actual measurement and chart |
| | Bottom characteristics | 2015.03 | Bottom characteristics of fishery port | Actual measurement |
| Data about fishery port facilities | Shoreline | \ | Elevation of shoreline | Actual measurement |
| | Seawall | \ | Elevation of seawall | Actual measurement |
| | Fishing-boat | \ | Length, width and draught | Actual measurement |
| | Anchor | \ | Weight and length | Actual measurement |

**2.3 Methods**
In this study, a framework (Fig. 2) is proposed for evaluating the resilience of fishery ports to typhoon related damage. The
framework is composed of the following five parts: typhoon building, model configuration and verification, hazard simulation,
individual assessment, and comprehensive assessment.
For typhoon building, for the convenience of reading, there are some terms that need to be explained first. The *influential*
*typhoons* refer to historical typhoons that have had an impact on the study area within a certain distance. The *typical typhoons*
are typhoon categories classified from the influential typhoons by certain rules. The *alternative assessment typhoons* refer to
alternative typhoon prototypes that are representative in each typical typhoon category. The final *assessment typhoon* is the
typhoon with the maximum destructive parameters among the alternative assessment typhoons.
Using MIKE21 software, the current, storm surge, and waves under various typhoon scenarios were simulated. These
scenarios provided the information required for the assessment and were chosen so that the data would cover future typhoon
events anticipated to have significant impacts on the Dongsha fishery port. The wind data and current data were used to
calculate the stresses on fishing boats, which were compared with the holding power of anchors. The resilience of the anchorage
to typhoon damage was represented by the minimum typhoon intensity when the force from the wind and currents was larger
than the holding power of the anchor. In a similar manner, the resilience of the shoreline facilities to typhoon damage was
represented by the minimum typhoon intensity when the water level of storm surge adding to 1/2 the significant wave height
was higher than the coastline elevation. Similarly, the resilience of the seawall to typhoon damage was represented by the
minimum typhoon intensity when the significant wave height or the sheltered area of the typhoon was higher than the design
wave of the seawall.

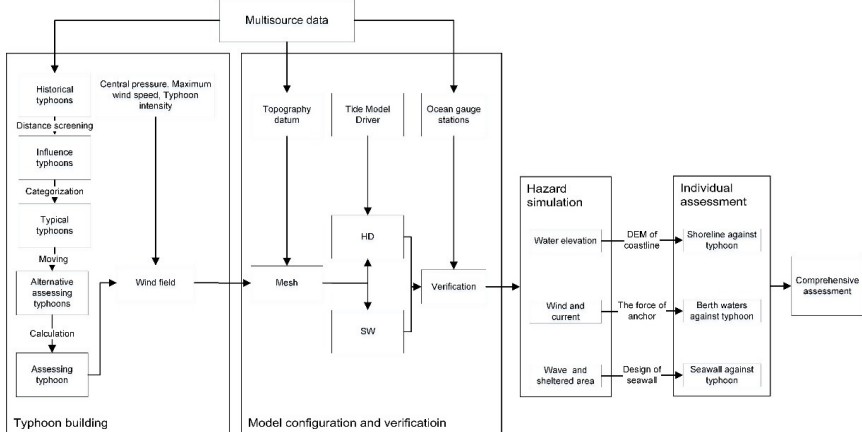

**Figure 2.** Framework for assessing the resilience of fishery ports to typhoon damage.
**2.3.1 Numerical model configuration**
The 2D shallow water model has been shown to reproduce storm surges well (Bertin et al., 2012). Notably, MIKE21 was used
successfully for the simulation of tidal waves during a storm surge in the north part of Liaodong Bay (Kong, 2014). In this
study, the MIKE21 model was used to construct the hydrodynamic module, storm surge, and typhoon waves. The model was
based on a flexible mesh approach. Simulations were made by using the SW model coupled with the HD FM model of the
software. The SW model solved for the wave action density, data which grew with the wind and dissipated owing to white
capping, surf breaking, bottom friction, and nonlinear interactions between spectral components in deep and shallow waters.
MIKE21 FM uses the finite volume method to solve the Navier–Stokes equations. Unstructured meshes were used in the model,
along with atmospheric pressure and wind. Detailed information for MIKE21 can be found in the scientific documentation and
user guide for the model (DHI, 2012).
The inset of Fig. 3 shows the computational domain and the mesh grid. It covered a large area that ranged from 106° to
135°E and 12° to 41°N; a large area was used to properly reproduce storm surges and waves generated at a greater distance
from the Dongsha fishery port. The grid used was fine near the area of interest and decreased in resolution in the deepwater



area where minute details were not as important. There were 67,549 grid cells and 35,899 nodes, which became denser closer
to the Dongsha fishery port. The minimum resolution of the grid size was 20 m, which could embody the seawall, wharf, and
other structures. The bathymetry data were obtained from several charts from the Maritime Safety Administration of the
People's Republic of China and actual measurements, which were unified at the same datum plane of the 1985 national height
datum.

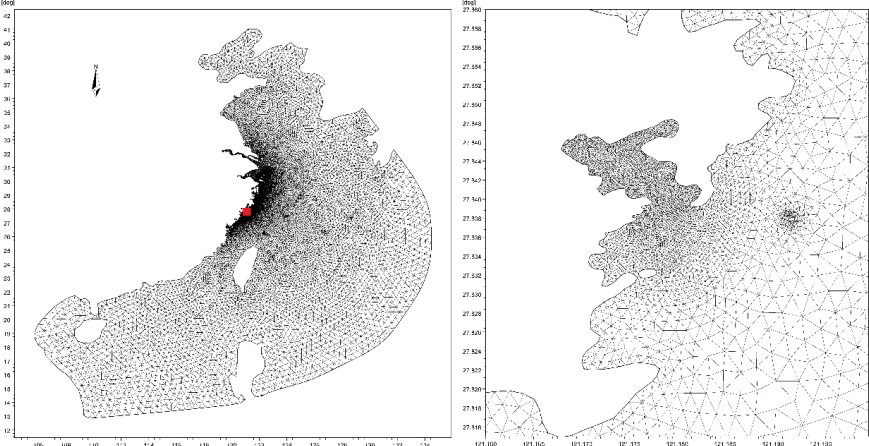


**Figure 3.** Mesh grid in the numerical model and grid of the interest area.

**2.3.2 Numerical model verification**
The typhoons of 9711, 0509, 0713, 0716, and 1509 were selected when the observed values were the maximum or the typhoon
caused relatively extensive damage. The numbers published by the China Meteorological Administration are indicative of the
year and order of typhoons that have impacted China, for example, 9711 means the 11[th] typhoon that occurred during 1997.
There were some ocean gauge stations near the Dongsha fishery port, and each station observed different oceanographic
elements. The storm surge model was validated with the data from the Kanmen and Wenzhou tide gauge stations. The wave
model was validated with the data from the Nanji and Wenzhou wave gauge stations. The whole hourly storm surge was
processed by simulations under various scenarios with and without a typhoon to extract the tide. To validate the surge, observed
and modeled water levels were compared. Figure 4 and 5 shows that there was a good correlation between the data from the
tide gauge stations and the model results, both in terms of the phase and amplitude. Because the maximum data were more
important during the assessment, Tables 2 and 3 respectively show the relative error of the maximum storm surge and waves.
The relative error of the maximum storm surge was 21.89 %, and that of the waves was 12 %. Modeling with good results very
similar to the observed data was very difficult to achieve, as the wind, rain, current, and wave interactions were complex during
a typhoon. However, the preliminary results showed that it was possible to forecast the effects of storm surges and waves by
several days in advance.

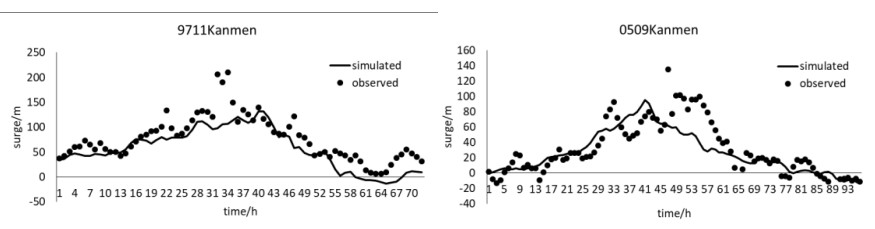

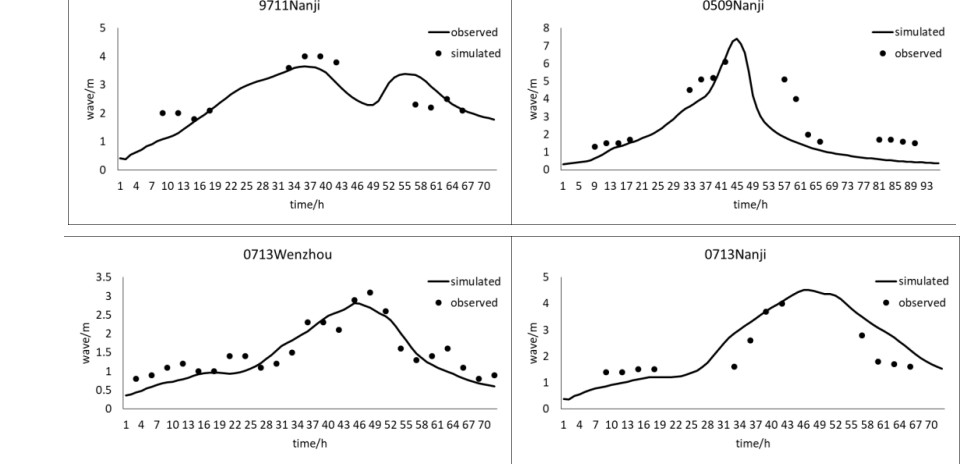

**Figure 4.** Comparison of the storm surge at Nanji and Wenzhou stations.





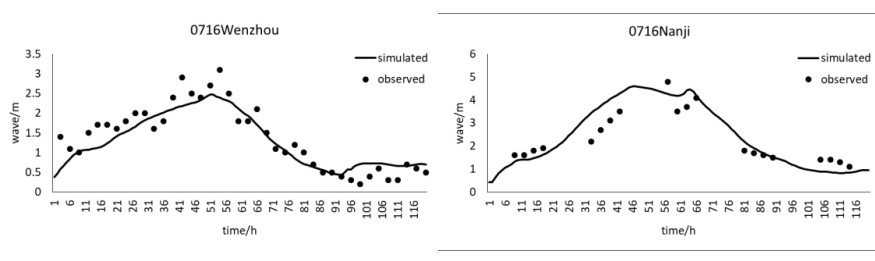


**Figure 5.** Comparison of the waves at Nanji and Wenzhou stations.

**Table 2.** Relative error of the storm surge.

| Typhoon | Station | Storm surge | | Relative error |
| --- | --- | --- | --- | --- |
| | | Observed (m) | Modeled (m) | |
| 9711 | Kanmen | 2.10 | 1.90 | 9.5 % |
| 0509 | Kanmen | 1.35 | 0.95 | 29.6 % |
| | Wenzhou | 1.08 | 0.99 | 8.3 % |
| 0713 | Kanmen | 1.02 | 0.76 | 25.5 % |
| | Wenzhou | 0.59 | 0.42 | 28.8 % |
| 0716 | Kanmen | 0.64 | 0.72 | 12.5 % |
| | Wenzhou | 0.53 | 0.76 | 43.4 % |
| 1509 | Kanmen | 1.43 | 1.02 | 28.7 % |
| | Wenzhou | 1.22 | 1.09 | 10.7 % |


**Table 3.** Relative error of the waves.

| Typhoon | Station | Wave | | Relative error |
| --- | --- | --- | --- | --- |
| | | Observed (m) | Modeled (m) | |
| 9711 | Nanji | 4 | 3.6 | 10 % |
| 0509 | Nanji | 6.1 | 7.4 | 21 % |
| 0713 | Nanji | 3.1 | 2.9 | 6 % |
| | Wenzhou | 4 | 4.5 | 13 % |
| 0716 | Nanji | 3.1 | 2.4 | 23 % |
| | Wenzhou | 4.8 | 4.6 | 4 % |
| 1509 | Wenzhou | 4.5 | 4.2 | 7 % |


**2.3.3 Typhoon prototype selection/ storm track**
Influence typhoons were chosen by a method of distance screening from the history of typhoons, which amounted to 1841
typhoons in total for China during the period 1949–2017. The method of distance screening involved drawing a circle with the
fishery port at the center and a radius of 40 km. This radius was set because the geometric mean radius of maximum wind is





47.5 km in the Atlantic and eastern Pacific (Willoughby and Rahn, 2004) and concentrated at 40 km in the western North
Pacific (Yang et al., 2017). The influential typhoons were classified in order to determine typical typhoon conditions.
Assessment of the typhoons was carried out with the maximum risk for alternative assessment typhoons according to the results
of simulations.

First, 1841 historical typhoon pathways were collected from the tropical cyclone information center of the China

Meteorological Administration, and these typhoons all occurred from 1949 to 2017. Next 123 influential typhoons were chosen
by the method of distance screening from the abovementioned historical typhoon pathways (Fig. 6). Because the influential
typhoons occurred in all directions, the influential typhoons were categorized into four typical typhoon patterns according to
their pathways as shown in Fig. 7. At the same time, by considering the opening direction of the Dongsha fishery port where
the seawall gap faces toward the southeast, the typhoon pathway toward the northwest was selected as the fifth typical typhoon
pattern. Then, five representatives were selected from each typical typhoon pattern. Next, five representatives were moved to
a radius of 40 km around the Dongsha fishery port, and these represented the alternative assessment typhoons (Fig. 8 and Table

4).

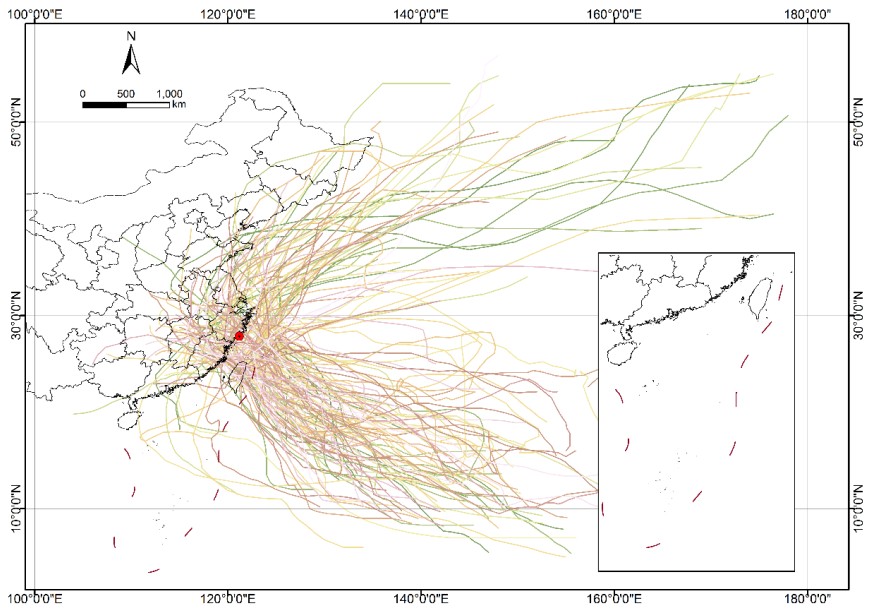


**Figure 6.** Pathways of influential typhoons.



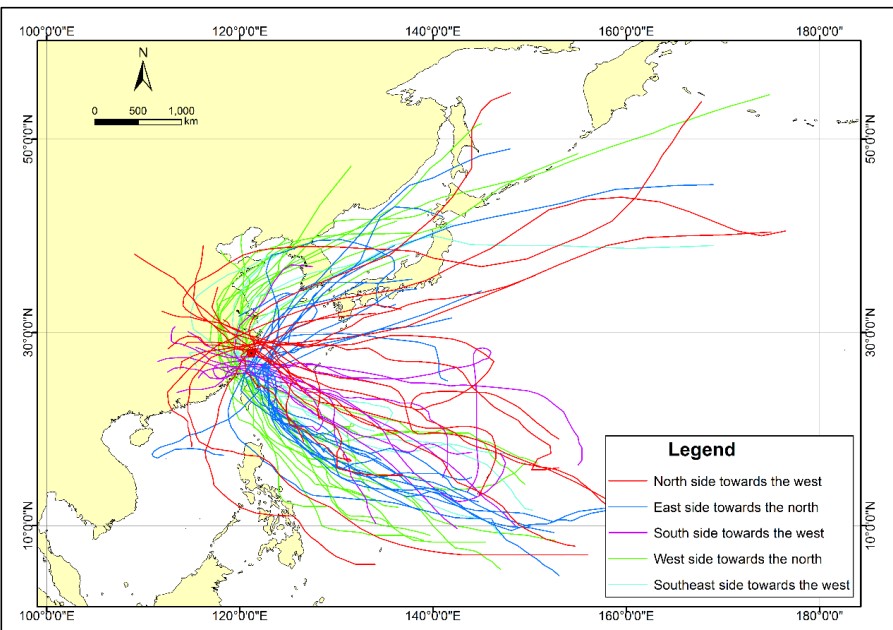

**Figure 7.** Pathways of typical typhoons.

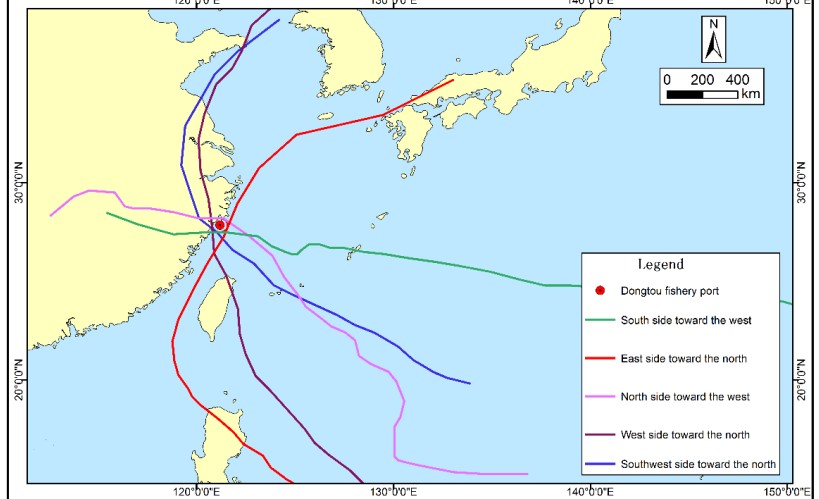

**Figure 8.** Pathways of alternative assessment typhoons.

**Table 4.** Pathways of alternative assessment typhoons.

| | Alternative assessment typhoons | Typhoon prototype number | Typhoon prototype name |
|---|---|---|---|
| 1 | East side toward the north | / | Virginia |
| 2 | West side toward the north | 8707 | Alex |
| 3 | South side toward the west | 0216 | Sinlaku |
| 4 | North side toward the west | 0414 | Rananim |





| 5 | Southeast side toward the west | 0713 | Wipha |
|---|---|---|---|


Five scenarios of different alternative assessment pathways under a level 17 typhoon were calculated, including the storm
surge and typhoon waves. Seven feature points, as shown in Fig. 9, were extracted from the results to reflect the area of the
seawall (B1), entrance (B2), anchorage water (B3, B4), wharf (B5, B6), and Dawangdian Bay (B7). The south side toward the
west scenario was selected as the final assessment typhoon pathway (Fig. 10), during which the storm surge and waves were
at the maximum values at the feature point (Table 5).

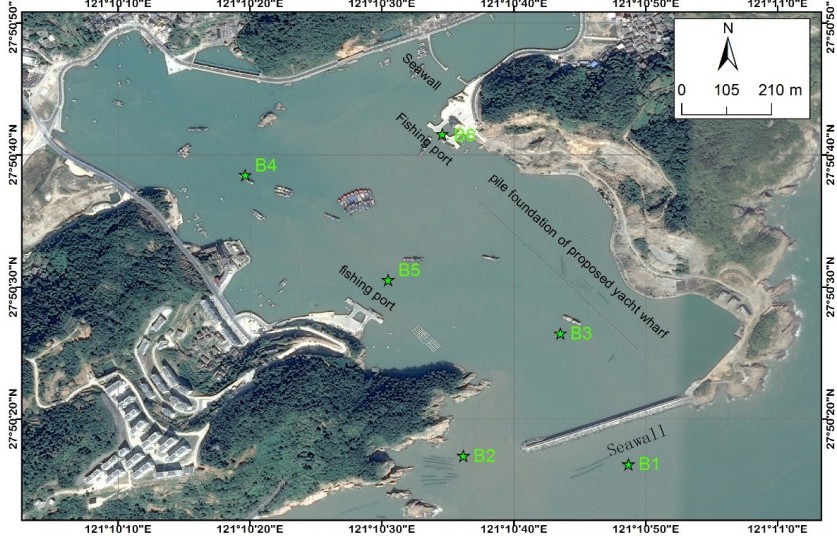


**Figure 9.** Feature points of the five alternative assessment typhoon scenarios (from © Google Earth).

**Table 5.** Results for the five typhoon pathways.

| Feature point | East side toward the north | | West side toward the north | | South side toward the west | | North side toward the west | | Southeast side toward the west | |
|---|---|---|---|---|---|---|---|---|---|---|
| | Surge | Wave | Surge | Wave | Surge | Wave | Surge | Wave | Surge | Wave |
| B1 | 0.72 | 3.49 | 0.57 | 2.74 | 2.03 | 3.61 | 1.70 | 1.57 | 0.80 | 3.14 |
| B2 | 0.72 | 3.12 | 0.57 | 2.55 | 2.04 | 3.34 | 1.71 | 1.57 | 0.81 | 2.76 |
| B3 | 0.72 | 0.31 | 0.57 | 0.22 | 2.03 | 0.28 | 1.71 | 0.10 | 0.81 | 0.27 |
| B4 | 0.73 | 0.21 | 0.58 | 0.16 | 2.06 | 0.21 | 1.74 | 0.08 | 0.82 | 0.15 |
| B5 | 0.73 | 0.11 | 0.57 | 0.09 | 2.05 | 0.12 | 1.73 | 0.04 | 0.82 | 0.08 |
| B6 | 0.73 | 0.35 | 0.57 | 0.26 | 2.04 | 0.34 | 1.73 | 0.12 | 0.81 | 0.27 |
| B7 | 0.73 | 0.14 | 0.57 | 0.11 | 2.04 | 0.14 | 1.73 | 0.05 | 0.81 | 0.11 |
| Mean | 0.73 | 1.10 | 0.57 | 0.88 | 2.04 | 1.15 | 1.72 | 0.50 | 0.81 | 0.97 |
| Maximum | 0.73 | 3.49 | 0.58 | 2.74 | 2.06 | 3.61 | 1.74 | 1.57 | 0.82 | 3.14 |



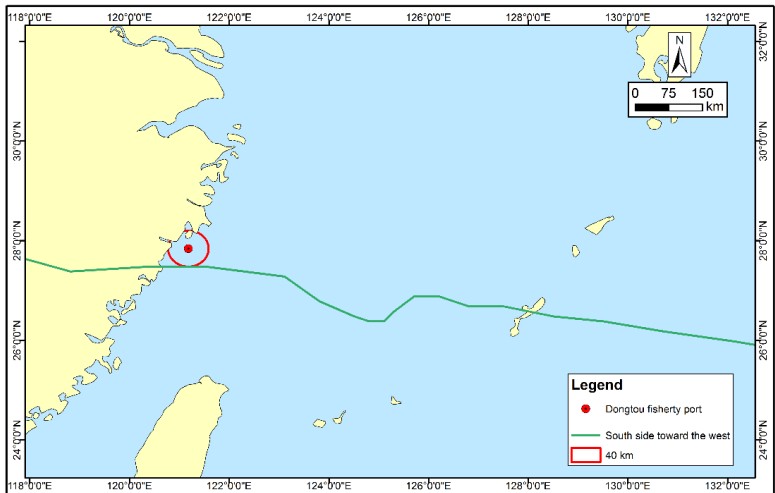

**Figure 10.** Pathway of the assessment typhoon.

In this study, the main approaches used for the typhoon wind field modeling were described by the Fujita Model, which has been employed in this area (Huang, 2017; Fujita, 1952).

**2.3.4 Parameter setting**

The tide of the open boundary was determined by using the tide obtained from the Tide Model Driver (TMD) package with its harmonic components (M2, S2, N2, K2, K1, O1, P1, Q1, and M4). The resulting forcing had a time step of 1 h. The input parameters for the wind model were the radius of maximum winds, traveling speed, and pressure difference between the storm's central pressure and the ambient (or peripheral) pressure. The radius of maximum winds was estimated from available observations by using a previously published empirical formula (Eq. (14). Zhu and Huang, 2002):

$$R = R_k - 0.4 \times (P_0 - 900) + 0.01 \times (P_0 - 900)^2 \tag{1}$$

where $R_k$ is the empirical parameter (usually a value of 40 km was used), $R$ is the radius of maximum winds, and $P_0$ is the central pressure.

The traveling speed for the forward velocity of the storm was obtained from the observed value of the prototype typhoon, for which data were collected from the China Meteorological Administration. The pressure difference of the typhoon was derived from the wind information provided in the typhoon history. According to experience and norms (China Meteorological Administration, 2006), 10 different values for the central pressure were used, namely, 995, 991, 985, 975, 965, 955, 945, 935, 925, and 915 (Table 6).

**Table 6**. Wind and pressure parameters of simulated typhoons.

| Typhoon level | 8 | 9 | 10 | 11 | 12 | 13 | 14 | 15 | 16 | 17 |
|---|---|---|---|---|---|---|---|---|---|---|
| Maximum wind speed (m s$^{-1}$) | 20 | 24 | 27 | 31 | 35 | 40 | 44 | 49 | 53 | 57 |
| Central pressure (hPa) | 995 | 991 | 985 | 975 | 965 | 955 | 945 | 935 | 925 | 915 |

**2.3.5 Assessment**

The forces exerted on fishing boats from wind were divided into lateral and vertical directions as follows:



$$F_{xw} = 73.6 \times 10^{-5} A_{xw} V_x^2 \zeta_1 \zeta_2 \qquad (2)$$
$$F_{yw} = 49.0 \times 10^{-5} A_{yw} V_y^2 \zeta_1 \zeta_2 \qquad (3)$$
where $F_{xw}$ and $F_{yw}$ are the component forces from wind in the lateral and vertical directions (kN), respectively; $A_{xw}$ and $A_{yw}$ are
the above water force area in the lateral and vertical directions (m²), respectively; $V_x$ and $V_y$ are the wind speed in the lateral
and vertical directions (m s⁻¹), respectively; $\zeta_1$ is a nonuniform coefficient that was set to the recommended value of 1 in this
study; and $\zeta_2$ is the altitude correction factor that was set to the recommend value of 1 in this study (Ministry of Transport of
the People's Republic of China, 2006).
The forces exerted on fishing boats from currents were calculated by the following formulas:
$$F_{xsc} = C_{xsc} \frac{\rho}{2} V_2 B' \qquad (4)$$
$$F_{ysc} = C_{ysc} \frac{\rho}{2} V_2 B' \qquad (5)$$
where $F_{xsc}$ and $F_{ysc}$ are the component forces from currents in the lateral and vertical directions (kN), respectively; $C_{xw}$ and $C_{yw}$
are the coefficients of the fore and aft, which were obtained from a look-up table (Ministry of Transport of the People's
Republic of China, 2006) as 0.09 and 0.04, respectively; $V$ is the current speed (m s⁻¹); $\rho$ is the water density (kg m⁻³); and $B'$
is the underwater area of the lateral direction (m²).
The force exerted on the ships was the resultant force of the wind and current:
$$\sum F = \sqrt{\left( \sum F_x \right)^2 + \left( \sum F_y \right)^2} \qquad (6)$$
The anchor holding power of fishing boats was calculated by the following formula:
$$P = P_a + P_c = \lambda_a W_a + \lambda_c W_c l \qquad (7)$$
where $P$ is the resultant force of anchor holding (kN); $P_a$ is the force of anchor holding (kN); $P_c$ is the force of anchor chain
holding (kN); $\lambda_a$ is the coefficient of the anchor, which was set to 3.5 in accordance with the clayey silt bottom material; $\lambda_c$ is
the coefficient of the anchor chain, which was set to 0.6 in accordance with the clayey silt bottom material; $W_a$ is the anchor
weight, which was set to 0.15 t, 0.5 t, and 0.7 t for large, medium, and small types of fishing boats; $W_c$ is the anchor chain
weight per meter; and $l$ is the length of the anchor chain underground. The resultant force of the fore and aft was 1.3 times the
resultant force.
**3 Results**
The water level is presumed to be a superposition of the tide and surge. The impacts of typhoon parameters on the storm were
studied (Wang et al., 2020). Storm surges are known to have some potential interactions with tides (Flather, 2001). Idier et al.
(2012) concluded that the instantaneous tide–surge interaction is non-negligible in the eastern half of the English Channel,
where it reaches values of 74 cm in the Dover Strait. From an operational perspective, an understanding of this interaction is
of value in order to choose relevant strategies in the risk analysis. Thus, to better assess the resistance level of the fishery port
against typhoon damage, tide–surge interactions were investigated.
The coupling processes of storm surges and tides were simulated in the following way. The surges were computed by
gradually adding 2 h tide interactions under the level 17 typhoon. Considering the tide period in this area, there were seven
scenarios. "ST-2" represented 2 h after the "ST" scenarios, and "ST+2" represented 2 h before the "ST" scenarios. In this study,
as shown in Table 7, the maximum storm surge occurred during the "ST-6" scenarios, that is, most of the largest practical storm
surges occurred around low tide, which is similar to results of the other study (Idier et al., 2012). Then, "ST-6" scenarios as


tide–surge interaction conditions were used for further simulation.

**Table 7.** Storm surge for different scenarios under a level 17 typhoon (cm).

| Scenario | Feature point | | | | | | |
|---|---|---|---|---|---|---|---|
| | B1 | B2 | B3 | B4 | B5 | B6 | B7 |
| ST-6 | 253 | 255 | 255 | 260 | 258 | 257 | 257 |
| ST-4 | 242 | 244 | 244 | 248 | 246 | 246 | 246 |
| ST-2 | 222 | 223 | 222 | 225 | 225 | 224 | 224 |
| ST | 203 | 204 | 203 | 206 | 205 | 204 | 204 |
| ST+2 | 207 | 208 | 208 | 211 | 210 | 209 | 209 |
| ST+4 | 218 | 220 | 219 | 223 | 222 | 221 | 221 |
| ST+6 | 229 | 231 | 231 | 235 | 233 | 233 | 233 |


Two types of runs were implemented with the HD model, namely, one with the forcing (tide, wind, atmospheric pressure)
and the other with the tide only. Based on historical storms and in collaboration with constructive typhoon characteristics, a
suit of typhoon scenarios under level 8–17 typhoons were created for surge and wave modeling using HD and SW. These
scenarios provided the information required for the assessment and were chosen so that the data would cover future typhoon
events anticipated to have significant impacts on the Dongsha fishery port.
Next the results will be analyzed considering the following three aspects: seawall, berth waters, and shoreline.
**3.1 Seawall**
The design and construction data for the seawall shows that the design wave elements $H_{1/3}$ of a 50-year return period is 6.5 m
at the seawall head, and the $H_{1/3}$ was 6.7 m at the seawall toe. The data extracted from typhoon scenario calculations were
compared with the design wave elements (Tables 8 and 9). The design wave elements are smaller than the calculated elements
at both the seawall head and seawall toe under a level 13 typhoon. Additionally, to resist a typhoon, the design wave elements
should be larger than the calculated elements. Thus, from the design wave point, the resistance level of the Dongsha fishing
against typhoon damage is 12.

**Table 8.** $H_s$ at seawall feature points under different typhoon levels.

| Typhoon level | $H_s$ at seawall head (m) | $H_s$ at seawall toe (m) |
|---|---|---|
| 8 | 3.5 | 3.6 |
| 9 | 4.2 | 4.2 |
| 10 | 5.3 | 5.1 |
| 11 | 6.2 | 6.1 |
| 12 | 6.4 | 6.3 |
| 13 | 7.1 | 7.0 |
| 14 | 7.3 | 7.3 |
| 15 | 7.7 | 7.8 |
| 16 | 8.3 | 8.2 |
| 17 | 8.3 | 8.5 |
| Design wave elements | 6.5 | 6.7 |


According to the design data, the sheltered areas for large, medium, and small types of fishing boats are 70,000 $m^2$, 280,000
$m^2$, and 180,000 $m^2$, respectively. Anchoring wave conditions of large, medium, and small types of fishing boats are 1.2 m,
1.0 m, and 0.5 m, respectively. A distribution map of the wave amplification that propagated into the port is shown in Fig. 11.




Because it is shielded by Dongtou Island, inrushing waves at the fishery port are small. The sheltered areas of level 8–17
typhoon scenarios are presented in Table 9. The areas where $H_{1/3}$ is smaller than 0.5 m, 1.0 m, and 1.2 m are compared between
the design and simulation. For instance, the design area where $H_{1/3} < 0.5$ m is $18 \times 10^4$ m², which is much smaller than the
simulated sheltered area $65.1 \times 10^4$ m² under the level 8 typhoon. Under the same typhoon level, the design area where $H_{1/3} <$
1.0 m is $(18+28) \times 10^4$ m², which is still smaller than the simulated sheltered area $(65.1+3.1) \times 10^4$ m². Similarly, the design
area where $H_{1/3} < 1.2$ m is $(18+28+7) \times 10^4$ m², which is also smaller than the simulated sheltered area $(65.1+3.1+0.2) \times 10^4$
m² under the level 8 typhoon. Thus, we could conclude that the Dongsha fishery port can resist the level 8 typhoon from the
aspect of the sheltered area. In a similar manner, the comparisons were carried out at the remaining typhoon levels. The results
showed that the maximum resistance level of the Dongsha fishery port against typhoon damage is 16.
According to the principle of high not low, the resistance level of Dongsha fishery port against typhoon damage is 12.

**Table 9.** Sheltered area under different typhoon levels.

| $H_{1/3}$ (m) | Design area ($\times 10^4$ m²) | Sheltered area under different typhoon levels ($\times 10^4$ m²) | | | | | | | | | |
|---|---|---|---|---|---|---|---|---|---|---|---|
| | | 8 | 9 | 10 | 11 | 12 | 13 | 14 | 15 | 16 | 17 |
| $H_{1/3} < 0.5$ | 18 | 65.1 | 62.2 | 53.5 | 48.3 | 44.9 | 41.6 | 37.7 | 34.1 | 35.1 | 0.0 |
| $0.5 < H_{1/3} < 1.0$ | 28 | 3.1 | 5.6 | 13.4 | 17.3 | 19.0 | 20.2 | 21.4 | 18.7 | 19.4 | 49.2 |
| $1.0 < H_{1/3} < 1.2$ | 7 | 0.2 | 0.3 | 0.6 | 1.4 | 2.3 | 3.3 | 4.5 | 7.2 | 6.8 | 5.3 |


Natural Hazards
and Earth System
**Figure 11.** Distribution maps of the waves under level 8–17 typhoons (from © Google Earth).

**3.2 Berth waters**

A total of 23 feature points were selected for fishing boats anchored in water in accordance with information from the fishery




port's administration department (Fig. 12). In Fig. 12, the rectangles represent berth waters and the feature points are at the
centers of the rectangles. Considering the long period force on fishing boats, the data for the wind and currents at those points
were extracted from a suit of typhoon scenarios under level 8–17 typhoons (Table 10).

**Table 10.** Force from wind and currents under different typhoon levels (kN).

| Feature point | Force under different typhoon levels | | | | | | | | | |
|---|---|---|---|---|---|---|---|---|---|---|
| | 8 | 9 | 10 | 11 | 12 | 13 | 14 | 15 | 16 | 17 |
| xx1 | 4.115 | 5.143 | 6.625 | 9.157 | 10.788 | 11.800 | 12.524 | 13.233 | 14.236 | 15.808 |
| xx2 | 4.116 | 5.144 | 6.627 | 9.161 | 10.794 | 11.805 | 12.529 | 13.238 | 14.239 | 15.812 |
| xx3 | 4.116 | 5.146 | 6.631 | 9.173 | 10.811 | 11.826 | 12.553 | 13.263 | 14.272 | 15.851 |
| xx4 | 4.115 | 5.140 | 6.618 | 9.147 | 10.778 | 11.788 | 12.512 | 13.220 | 14.224 | 15.798 |
| xx5 | 4.133 | 5.165 | 6.649 | 9.180 | 10.810 | 11.822 | 12.548 | 13.257 | 14.261 | 15.834 |
| xx6 | 4.115 | 5.140 | 6.618 | 9.148 | 10.778 | 11.788 | 12.512 | 13.220 | 14.223 | 15.796 |
| xx7 | 4.107 | 5.134 | 6.614 | 9.145 | 10.776 | 11.787 | 12.511 | 13.220 | 14.222 | 15.795 |
| xx8 | 4.107 | 5.132 | 6.610 | 9.140 | 10.770 | 11.781 | 12.505 | 13.213 | 14.216 | 15.789 |
| xx9 | 4.106 | 5.131 | 6.609 | 9.139 | 10.769 | 11.779 | 12.503 | 13.212 | 14.215 | 15.787 |
| zx1 | 10.769 | 13.458 | 17.325 | 23.950 | 28.223 | 30.871 | 32.769 | 34.628 | 37.260 | 41.384 |
| zx2 | 10.814 | 13.498 | 17.367 | 23.988 | 28.255 | 30.899 | 32.794 | 34.649 | 37.274 | 41.390 |
| zx3 | 10.761 | 13.445 | 17.317 | 23.944 | 28.214 | 30.860 | 32.756 | 34.610 | 37.235 | 41.352 |
| zx4 | 10.786 | 13.478 | 17.352 | 23.978 | 28.248 | 30.893 | 32.788 | 34.643 | 37.265 | 41.382 |
| dx1 | 13.767 | 17.213 | 22.199 | 30.715 | 36.199 | 39.614 | 42.074 | 44.469 | 47.850 | 53.162 |
| dx2 | 13.844 | 17.255 | 22.180 | 30.665 | 36.145 | 39.539 | 41.980 | 44.354 | 47.695 | 52.964 |
| dx3 | 13.933 | 17.343 | 22.268 | 30.693 | 36.121 | 39.479 | 41.895 | 44.251 | 47.589 | 52.830 |
| dx4 | 14.274 | 17.681 | 22.602 | 31.025 | 36.451 | 39.808 | 42.225 | 44.580 | 47.918 | 53.159 |
| dx5 | 13.843 | 17.254 | 22.179 | 30.605 | 36.033 | 39.391 | 41.811 | 44.172 | 47.501 | 52.742 |
| dx6 | 13.712 | 17.134 | 22.067 | 30.494 | 35.922 | 39.282 | 41.697 | 44.048 | 47.387 | 52.633 |
| dx7 | 13.701 | 17.112 | 22.038 | 30.464 | 35.892 | 39.250 | 41.667 | 44.023 | 47.361 | 52.602 |
| dx8 | 13.702 | 17.119 | 22.052 | 30.480 | 35.908 | 39.264 | 41.679 | 44.034 | 47.373 | 52.616 |
| dx9 | 13.705 | 17.116 | 22.042 | 30.468 | 35.896 | 39.254 | 41.671 | 44.027 | 47.365 | 52.606 |
| dx10 | 14.301 | 17.708 | 22.629 | 31.051 | 36.477 | 39.834 | 42.250 | 44.606 | 47.944 | 53.184 |


In the Dongsha fishery port, each boat is anchored by two anchors on the fore and aft. The forces of fore and aft are
considered. By comparing the force exerted on the ship with the resultant force of the fore and aft (Table 11), it could be
concluded that the resistance level of the Dongsha fishing against typhoon damage is 12.

**Table 11.** Force of anchor holding (kN).

| Ship type | Force of anchor | Force of anchor chain | Resultant force | Resultant force of fore and aft |
|---|---|---|---|---|
| Small | 5.145 | 4.704 | 9.849 | 12.804 |
| Medium | 17.15 | 4.704 | 21.854 | 28.410 |
| Large | 24.01 | 4.704 | 28.714 | 37.328 |






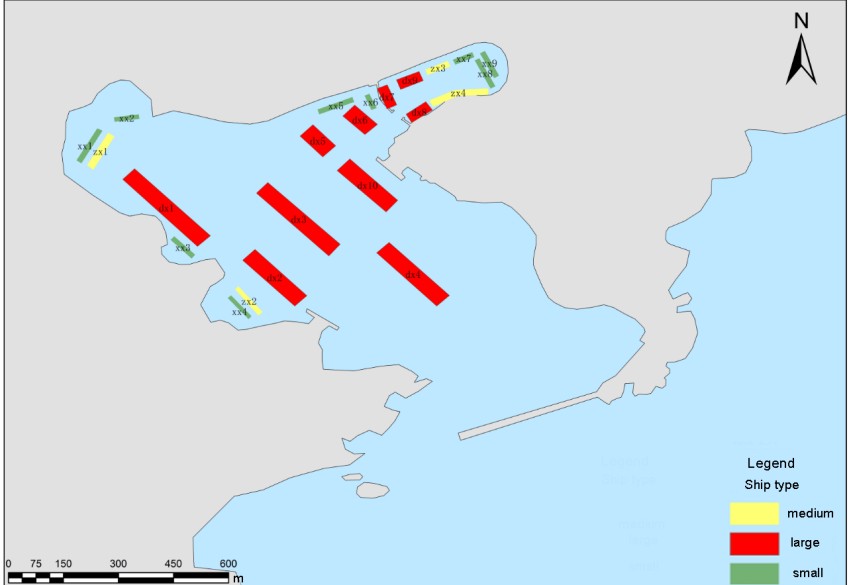

**Figure 12.** Feature points of the fishing boat anchoring water.

**3.3 Shoreline**

In consideration of the features of the Dongsha fishery port, 20 points were selected to represent the different types of shoreline (Fig. 13). Regarding the typhoon rating assessment for the shoreline, knowledge on the elevation of the coastline and the water was required. The water elevation was the height of the storm surge adding to 1/2 $H_s$. The results for the shoreline resilience to typhoons are shown in Table 12 and Fig. 13. The elevation of the shoreline should be higher than that of the water. Therefore, it could be concluded that the resistance level of Dongsha fishing against typhoon damage is 12.

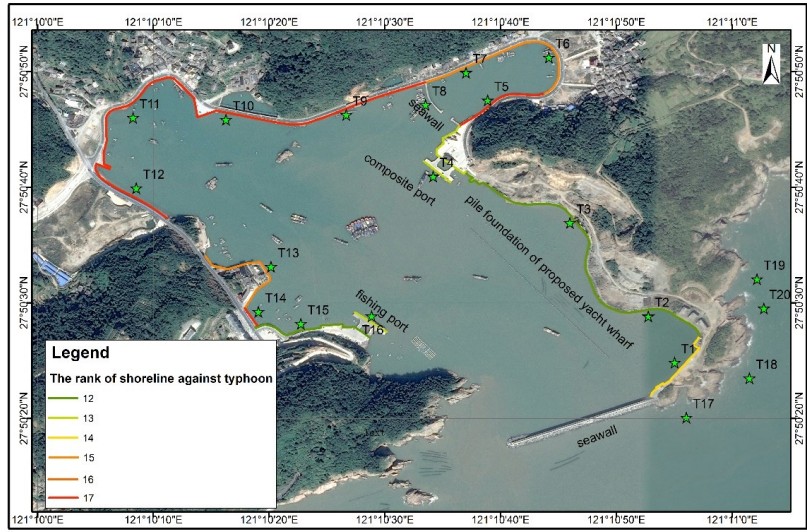

**Figure 13.** Feature points of different typhoon pathways scenarios (from ⓒ Google Earth).

**Table 12.** Water elevation and coastline elevation.

| Point | Water elevation under different typhoon levels (m) | | Coastline | Level |
|---|---|---|---|---|




| | 8 | 9 | 10 | 11 | 12 | 13 | 14 | 15 | 16 | 17 | elevation (m) | result |
|---|---|---|---|---|---|---|---|---|---|---|---|---|
| T1 | 3.02 | 3.22 | 3.51 | 3.91 | 4.20 | 4.42 | 4.56 | 4.77 | 4.93 | 5.27 | 4.57 | 14 |
| T2 | 3.05 | 3.26 | 3.56 | 3.98 | 4.27 | 4.51 | 4.67 | 4.87 | 5.02 | 5.37 | 4.43 | 12 |
| T3 | 3.12 | 3.34 | 3.67 | 4.10 | 4.40 | 4.63 | 4.80 | 4.96 | 5.11 | 5.48 | 4.53 | 12 |
| T4 | 3.10 | 3.30 | 3.59 | 4.00 | 4.29 | 4.50 | 4.65 | 4.81 | 4.96 | 5.26 | 4.53 | 13 |
| T5 | 3.00 | 3.18 | 3.44 | 3.81 | 4.09 | 4.27 | 4.40 | 4.53 | 4.70 | 4.96 | 5.20 | 17 |
| T6 | 2.99 | 3.17 | 3.43 | 3.79 | 4.06 | 4.24 | 4.37 | 4.50 | 4.67 | 4.92 | 4.52 | 15 |
| T7 | 3.01 | 3.19 | 3.46 | 3.83 | 4.10 | 4.29 | 4.43 | 4.56 | 4.72 | 4.99 | 4.68 | 15 |
| T8 | 3.03 | 3.23 | 3.50 | 3.89 | 4.16 | 4.36 | 4.50 | 4.63 | 4.79 | 5.06 | 4.17 | 12 |
| T9 | 3.10 | 3.30 | 3.59 | 3.98 | 4.27 | 4.47 | 4.61 | 4.75 | 4.91 | 5.20 | 5.21 | 17 |
| T10 | 3.07 | 3.27 | 3.55 | 3.95 | 4.24 | 4.44 | 4.59 | 4.72 | 4.88 | 5.17 | 5.27 | 17 |
| T11 | 3.00 | 3.15 | 3.38 | 3.64 | 3.80 | 3.90 | 4.01 | 4.12 | 4.26 | 4.48 | 5.27 | 17 |
| T12 | 3.04 | 3.24 | 3.51 | 3.84 | 4.07 | 4.23 | 4.31 | 4.43 | 4.59 | 4.84 | 5.56 | 17 |
| T13 | 3.04 | 3.23 | 3.51 | 3.90 | 4.18 | 4.38 | 4.52 | 4.66 | 4.82 | 5.10 | 4.81 | 15 |
| T14 | 3.01 | 3.21 | 3.48 | 3.87 | 4.15 | 4.34 | 4.48 | 4.62 | 4.77 | 5.05 | 5.36 | 17 |
| T15 | 3.01 | 3.21 | 3.47 | 3.86 | 4.13 | 4.32 | 4.46 | 4.60 | 4.77 | 5.04 | 4.17 | 12 |
| T16 | 3.05 | 3.24 | 3.53 | 3.92 | 4.20 | 4.41 | 4.56 | 4.71 | 4.87 | 5.15 | 4.56 | 13 |
| T17 | 4.75 | 5.24 | 5.97 | 6.70 | 7.17 | 7.72 | 8.01 | 8.31 | 8.20 | 8.93 | 9.21 | 17 |
| T18 | 4.77 | 5.25 | 5.95 | 6.72 | 7.19 | 7.78 | 8.10 | 8.38 | 8.25 | 8.89 | 12.23 | 17 |
| T19 | 4.86 | 5.33 | 6.00 | 6.81 | 7.22 | 7.79 | 8.11 | 8.40 | 8.29 | 8.82 | 9.13 | 17 |
| T20 | 4.83 | 5.31 | 5.98 | 6.77 | 7.20 | 7.79 | 8.10 | 8.38 | 8.26 | 8.80 | 9.13 | 17 |

### 3.4 Comprehensive assessment

According to the "Regulation for typhoon prevention assessment of fishery ports," there are two types of typhoon damage resistance levels for fishery ports. One represents the lowest level, while the other represents the comprehensive level. The lowest level of a fishery port represents the lowest values for the seawall, berth waters, and shoreline level, and the Dongsha fishery port was found to have a value of 12. The comprehensive level represents the weighted average of the seawall, berth waters, and shoreline level. The weighting factors of the seawall, berth waters, and shoreline are 0.25, 0.45, and 0.3, respectively. Hence, the calculated comprehensive level of the Dongsha fishery port is 12.

### 4 Discussion

The method introduced in this study is a practical technique for quantitatively assessing a fishery port's resilience to typhoon-related damage, and results are based on seawall, berth waters, and shoreline perspectives. Such an assessment of the resistance level of a fishery port against typhoon damage can reveal weaknesses in the port's defenses and allow for optimization of shelter spaces for fishing boats. The analysis carried out here had several caveats, which are important to highlight when considering these results. Notably, the level 12 for the Dongsha fishery port does not indicate that boats should be evacuated when a level 12 level typhoon is coming. However, when a level 12 typhoon slams into the Dongsha fishery port at the radius of maximum winds, boats should consider taking shelter. The feature points of T2, T3, T8, and T15 are the weaknesses of the Dongsha fishery port, and the port could enhance its defenses through increasing the elevation at these weakness points.

Considering the uniform standard, the analysis treated the distance from the fishery port to the storm track very roughly, which is the geometric mean radius of maximum wind. The other distance was also not taken into account in the assessment. In this analysis, all other impacts (sea level rise, rain, stability of infrastructure) were disregarded; the proposed methodology does not assess the total conditions of the fishery port.





In addition to the assessment of the resilience of the fishery port to typhoons, the weather forecasting and warning systems
established in Wenzhou have proven to be efficient at preventing human and economic losses from typhoons. Further,
evacuation plans and disaster response and preparedness solutions should be employed.
**5 Conclusion**
Some of the damage to fishery ports from typhoons may be preventable. This study described a systematic and quantitative
method for assessing the resilience of fishery ports to typhoons, and a case study was carried out on the Dongsha fishery port
in Zhejiang Province, China. Historical typhoons were studied to identify the most useful typhoon pathways (south side toward
the west) and scenarios (level 8–17 typhoons) for the assessment. The findings indicated that tide–surge interactions results,
albeit these data were based on a limited number of events, are important to consider and the majority of the largest practical
storm surges occurred around low tide; this was similar to the results in another study  (Idier et al., 2012). Importantly, the
Dongsha fishery port was found to have a resistance level of 12, and several points of weakness were identified where
improvements in elevation could lessen impacts from future typhoons. In conclusion, the findings of this study demonstrated
that this is a versatile framework for assessing fishing ports and developing disaster prevention plans. Though there remain a
few constraints in its application (such as with regard to sea level rise, rain, and the stability of infrastructure), the proposed
method should be readily applicable to other locations.
**Author contribution** Yachao Zhang and Jufei Qiu designed the versatile methodology for evaluation of the resilience of
fishery ports to typhoons. Yachao Zhang and Xiaojie Zhang prepared the manuscript with contributions from all co-authors.
Aifeng Tao, Jianli Zhao, Jianfeng Wang developed the model and performed the simulations. Yanfen Deng figures and Wentao
Huang analysed the result.
**Acknowledgments** We are grateful to the Wenzhou Marine Environmental Monitoring Center and Dongtou Fishery
Administration and Port Supervision station for providing basic data. This work was jointly supported by the Open Funds
program of the Key Laboratory of Coastal Disaster and Defense, Ministry of Education (No. 201604), Key Laboratory of
Integrated Marine Monitoring and Applied Technologies for Harmful Algal Blooms, S.O.A., MATHAB (No.
MATHAB201804) and Youth Marine Science Foundation of East China Sea Bureau of  Ministry of Natural Resources (No.

202002).

**Declarations**
**Competing interests**
Not applicable.
**Availability of data**
The datasets used during the current study are available from the corresponding author on reasonable request.
**Code availability**
Not applicable.

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
