# Peer review of "Evaluation of the resilience of fishery ports to typhoons: a case study on Dongsha fishery port"

_Natural Hazards and Earth System Sciences, 2021_

## Author Comment (AC1)

This study applied numerical simulation model to resolved tides and typhoon induced storm surge, waves in Dongsha fishery port, Zhejiang Province, China, the methodology can be used to acquire valuable information on the resilience of fishery ports to typhoons.

**Response:**

Thanks for your comments. We really appreciate your kind help in reviewing the manuscript. Detailed point-by-point responses are presented as below according to your comments.

Some technical problems are:

■ In model verification part, (1) several validation stations are applied, please mark these stations on the map, and list the location (longitude and latitude) in a table, explain the deviations of simulated results in detail. (2) What kind of driven winds are used to drive the model, only typhoon wind? (3) How about the boundary tidal elevation and storm surge elevation?

> **Response:**
>
> (1) In model verification part, the validation stations applied will be marked on map, and the location (longitude and latitude) will be listed in a table. For the deviations of simulated results, modeling results very close to the observed data are very difficult to achieve, as the wind, rain, current, and wave interactions are complex during a typhoon. Therefore when the average deviations are not bigger than 25% and the trends of simulated results are similar to the observed results, further steps can be taken to forecast the effects of storm surges and waves.
>
> (2) The driven winds were comprised of typhoon wind and background wind field.
>
> (3) The boundary tidal elevation and storm surge elevation were the measured data from ocean stations.

■ 2 hourly tide interactions with storm surge seems not enough, under a semidiurnal tidal situations, 1 hourly should be more accurate.

> **Response:**
>
> We agree that the tide-surge interactions will be more accurate with higher time accuracy. However, considering the amount of computation and computing power, we picked the 2 hourly tide interactions which was enough to reflect the tide-surge interactions in this study case.

■ In 3.1 Seawall section, please give the details of model setting of storm wave simulation.

> **Response:**
>
> The details of model setting of storm wave simulation will be given.

■ In 3.2 Berth waters section, please present methodology of how to calculate the force at the 23 feature points in the fishery port.

> **Response:**
>
> In the 2.3.4 section, there is the methodology of how to calculate the force at the 23 feature points in the fishery port.

Some minor problems:

■ In line 299, what does "Considering the long period force on fishing boats" mean? Which content in the text show the long period?

  **Response:**
  In general, the wind and current are longer period force on fishing boats than the wave. So "the long period force on fishing boats" here refers to the forces generated by wind and current.

■ In line 315, saying of "The water elevation was the height of the storm surge adding to 1/2 Hs" is not reasonable.

  **Response:**
  This saying may be not inappropriate. Because the interactions of storm surge and wave are complicated. This is just a simple and convenient method to calculate the water elevation when the interactions are not figured out.

■ Line 97, MIKE21 is not software.

  **Response:**
  The word "software" will be deleted.

■ Please show the model mesh of Dongsha fishery port zone.

  **Response:**
  The paper will show the model mesh of Dongsha fishery port zone.

---

## Author Comment (AC2)

The paper presents a methodology to assess the impact of typhoons on fishery ports, with a specific application to Dongsha. The authors make a determination of the worst typhoon that produces a major impact on the port, and from which, through the use of numerical modeling, they determine the degree of typhoon that would exceed the stability thresholds of the port defense structures, the level of flooding of the surrounding area, the breakage of fishing boat moorings and anchorage areas.

The paper is well structured and the subject matter is of great interest to the scientific community.

**Response:** Thanks for your comments. We really appreciate your kind help in reviewing the manuscript.

In general, I consider that the paper presents a methodology that is very simple and whose application introduces much uncertainty in the estimation of the impacts of typhoons on the coast, and more specifically in fishing ports. The method is purely deterministic in the analysis performed, lagging far behind other methods used and present in the literature for the analysis of typhoon trajectories and their impact on the coast. The method is too simplified, and the determination of the degree of the typhoon for the analysis of the impacts on the coast is not very accurate. The deterministic degree of the approach followed is one of the weakest points of the work and far from the state of the art on this subject.

**Response:** Thank you for your comments. First of all, this paper aims at proposing a systematic and quantitative method for assessing the resilience of fishery ports to typhoons. And we presented an integrated solution in the scenarios of typhoons. It is the overall technical framework that we focus on. Although some specific methods used seem simple, they are very easy to implement and have practical significance. Different form theoretical study on the analysis of typhoon trajectories, we focus on the typicality of a typhoon to a fishery port and whether the method can be replicated by different fishery ports. The determination of the degree of the typhoon is not very accurate as you said, because of the complex interactions of wind, rain, tide, current, and wave, which is still an academic challenge on present. After deriving a quantitative value for the resistance level of a fishery port against typhoons, even if it is not very accurate and is just a relative value, effective countermeasures for typhoons can be proposed. And such results would also be useful for administrators to make judgments on whether evacuations to a relatively safer port are needed for ships.

Another aspect that I would like to highlight is that both the text and the title speak of resilience. However, the paper does not perform a resilience analysis. The methodology presented is a pure methodology for analyzing the impacts of typhoons on the coast. It does not talk about the resilience of the port or its instigations, but only an analysis of the exceedance of a threshold in different impacts on the coast, due to the effect of exceeding a value of wave height or level. I ask the authors to remove the word resilience from the title and its references in the text.

**Response:** The "resilience" here means the resistance level of a fishery port against typhoons which is represented by the fishery port's maximum bearable level of typhoon. This paper does not study the physical property or elastic deformation of a fishery port against typhoons, but mainly considers dragging of anchor, design defense conditions of seawall, and the elevation of the coast from a risk perspective.

Regarding the assessment of impacts, I believe that the authors are using very simplified methods and do not detail very well other important processes. For example, in the seawall analysis, they use the exceedance of the design wave height of the structure to determine a failure of the structure. This is not entirely correct, since the stability of the structure can be compromised not only by the height value of the structure, but also by the water level. Failure can occur for a lower wave height value, with a higher water level value, as is possible as a consequence of the water level rise caused by typhoons. In the case of the coastline, I believe that the use of half the wave height for the calculation of the wave run-up, and subsequent assessment of flooding, is very rough. There are other methodologies based on the determination of the wave run-up, which do not require a higher computational cost, and allow a better estimation of this impact. Regarding the other two impacts, the anchorage area and the resistance of the moorings, I consider that the approximation used is correct.

**Response:** As mentioned above, this paper does not study the physical property or elastic deformation of a fishery port (i.e. a structure) against typhoons. And when a typhoon reaches the determined level, it does not mean the failure of the structure. In fact, we do not discuss the failure of a structure. We discuss the resistance of the seawall by comparing its design wave height and design sheltered area with those triggered by the interaction of seawall and typhoons on different level. And further study is needed to figure out whether the other methodologies based on wave run-up will allow a better estimation.

Regarding the hydrodynamic processes studied, I consider that the validation is not quite optimal, although the authors think so. I believe that the results show a clear underestimation of the storm surge elevation at the moments of maximum intensity in almost all the stations used to validate the model. This is key and very relevant in this case, since the methodology is based precisely on the worst case scenario and the maximum over-elevation, which is precisely when the storm surge value is most underestimated, and as a consequence, the impact.

**Response:** For the validation, modeling results very close to the observed data are very difficult to achieve, as the wind, rain, current, and wave interactions are complex during a typhoon. In figure 4, among 9 stations, four are underestimated, one are overestimated, and the rest four are well-estimated. Therefore when the average deviations are not bigger than 25% and the trends of simulated results are similar to the observed results, further steps can be taken to forecast the effects of storm surges and waves.

There is an additional aspect that is not discussed in detail and that is relevant, and that is the depth of study and the limitation of the wave height by depth. No data is given as to what is the depth in front of the seawall (I would use the word breakwater, instead of seawall; this is used for an element attached to the shore) and whether wave breaking affects the structure. The same comment applies to the characterization of wave run-up on shore. Another important aspect is that it is not detailed which wave height statistic is used. In the figures and in the text "wave" is mentioned, but it is not known if it is a significant wave height. Please detail.

**Response:** The depth in front of the seawall will be added. We did not consider the stability of the seawall, so we did not discuss whether wave breaking affects the structure. In this paper, the wave height statistic is the significant wave height represented by "$Hs$".

For all these comments I consider that the paper should be rejected for publication in the journal. First of all, the paper does not present a methodology for the analysis of resilience, but for the impact on the coast. I consider that the methodology presented is neither novel nor meaningful for impact calculation and that it presents many uncertainties in the results that make the impact assessment inadequate. In general, the work is far from state of the art.

**Response:** This paper proposed a systematic and quantitative method for assessing the resilience of fishery ports to typhoons. As mentioned above, the subject matter is of great interest to the scientific community. The methodology applied in this study is novel and provides an approach that can be applied universally to evaluate the resistance of various ports against typhoons or hurricanes. The overall technical framework provides a foundation for further studies on evaluation of the resilience of fishery ports to typhoons. And the assessment results would be useful for administrators to make judgments on whether evacuations to a relatively safer port are needed for ships and human beings. We believe that our study makes a significant contribution to the literature.